# Aerosol Surface Area Concentration: a Governing Factor for New Particle Formation in Beijing

Runlong Cai[1,#], Dongsen Yang[2,#], Yueyun Fu[1], Xing Wang[2], Xiaoxiao Li[1], Yan Ma[2], Jiming Hao[1], Jun Zheng[2*] and Jingkun Jiang[1*]

[1]State Key Joint Laboratory of Environment Simulation and Pollution Control, School of Environment, Tsinghua University, Beijing, 100084, China

[2]Collaborative Innovation Center of Atmospheric Environment and Equipment Technology, Nanjing University of Information Science & Technology, Nanjing 210044, China

#: Runlong Cai and Dongsen Yang contribute equally to this work

*: *Correspondence to*: J. Jiang (jiangjk@tsinghua.edu.cn) and J. Zheng (zheng.jun@nuist.edu.cn)

**Abstract.** The predominating role of aerosol Fuchs surface area, $A_{Fuchs}$, in determining the occurrence of new particle formation (NPF) events in Beijing was elucidated in this study. The analysis was based on a field campaign from 12 March 2016 to 6 April 6 2016 in Beijing, during which aerosol size distributions down to ~1 nm and sulfuric acid concentrations were simultaneously monitored. The 26 days were classified into 11 typical NPF days, 2 undefined days, and 13 non-event days. A dimensionless factor, $L_\Gamma$, characterized by the relative ratio of the coagulation scavenging rate over the condensational growth rate (Kuang et al., 2010), was applied in this work to reveal the governing factors for NPF events in Beijing. The three parameters determining $L_\Gamma$ are sulfuric acid concentration, the growth enhancement factor characterized by contribution of other gaseous precursors to particle growth, $\Gamma$, and $A_{Fuchs}$. Different from other atmospheric environments, such as in Boulder and Hyytiälä, the daily-maximum sulfuric acid concentration and $\Gamma$ in Beijing varied in a narrow range with geometric standard deviations of 1.40 and 1.31, respectively. A positive correlation between the estimated new particle formation rate, $J_{1.5}$, and sulfuric acid concentration was found with a mean fitted exponent of 2.4. However, the maximum sulfuric acid concentrations on NPF days were not significantly higher (even lower, sometime) than those on non-event days, indicating that the abundance of sulfuric acid in Beijing was high enough to initiate nucleation, but may not necessarily lead into NPF events. Instead, $A_{Fuchs}$ in Beijing varied greatly among days with a geometric standard deviation of 2.56, whereas the variabilities of $A_{Fuchs}$ in Tecamac, Atlanta, and Boulder were reported to be much smaller. In addition, there was a good correlation between $A_{Fuchs}$ and $L_\Gamma$ in Beijing ($R^2 = 0.88$). Therefore, it was $A_{Fuchs}$ that fundamentally determined the occurrence of NPF events. Among 11 observed NPF events, 10 events occurred when $A_{Fuchs}$ was smaller than 200 $\mu m^2/cm^3$. NPF events were suppressed due to the coagulation scavenging when $A_{Fuchs}$ was greater than 200 $\mu m^2/cm^3$. Measured $A_{Fuchs}$ in Beijing was in a good correlation with its $PM_{2.5}$ mass concentration ($R^2 = 0.85$) since $A_{Fuchs}$ in Beijing was mainly determined by particles in the size range of 50 – 500 nm that also contribute to the $PM_{2.5}$ mass concentration.

## 1 Introduction

New particle formation (NPF) is closely related to atmospheric environment. It is a common atmospheric phenomenon, which has been observed all over the world (Kulmala et al., 2004). High concentrations of ultrafine particles are formed intensively during NPF events. It has been illustrated through both theoretical modelling and field observations that these ultrafine particles can grow and serve as cloud condensation nuclei (Kuang et al., 2009; Spracklen et al., 2008), and thus affect climate (IPCC, 2013). The increased number concentration of ultrafine particles also raises concerns on human health (HEI, 2013).

New particles are formed by nucleation from gaseous precursors, such as sulfuric acid, ammonia, and organics. Newly formed particles either grow by condensation or are lost by coagulation with other particles (McMurry, 1983). Aerosol Fuchs surface area, $A_{\text{Fuchs}}$, is a parameter that describes the coagulation scavenging effect quantitatively. In addition to gaseous precursors participating in nucleation and subsequent condensational growth, it has been a consensus that the occurrence of a NPF event is also limited by $A_{\text{Fuchs}}$, because the survival possibility of nucleated particles is suppressed when the coagulation scavenging effect is significant (Weber et al., 1997; Kerminen et al., 2001; Kuang et al., 2012). Reported average $A_{\text{Fuchs}}$ (or in the form of condensation sink) on NPF days was found to be lower than that on non-event days at several locations (Dal Maso et al., 2005; Gong et al., 2010; Qi et al., 2015).

A dimensionless criterion, $L_{\Gamma}$, was proposed to characterize the ratio of particle scavenging loss rate over condensational growth rate, and to predict the occurrence of NPF events in diverse atmospheric environments (Kuang et al., 2010). By definition, $L_{\Gamma}$ is determined by three factors, i.e., the sulfuric acid concentration, the growth enhancement factor representing contribution of other gaseous precursors in addition to the sulfuric acid concentration, $\Gamma$, and $A_{\text{Fuchs}}$. The diurnal sulfuric acid concentration can vary drastically diurnal variation due to the substantial change in radiation (e.g., from several thousand to ~$1.5 \times 10^6$ #/cm$^3$ in this campaign) and the increase in sulfuric acid concentration after the sunrise can potentially lead to nucleation. The values of $A_{\text{Fuchs}}$, however, were usually reported within a narrow range at locations, such as Tecamac, Atlanta and Boulder (Kuang et al., 2010). The sulfuric acid concentration in Atlanta and Hyytiälä can differ significantly among days (Eisele et al., 2006; Petäjä et al., 2009). Therefore, the sulfuric acid concentration often governs nucleation and subsequent growth in the sulfur-rich atmosphere, such as in Atlanta (McMurry et al., 2005). The growth enhancement factor, $\Gamma$, at Hyytiälä varied in a wide range while those at Tecamac and Boulder were found in a relatively narrow range.

Aerosol concentrations in Beijing are usually much higher than those in clean environments. The annual average PM$_{2.5}$ mass concentration in 2016 was 73 μg/m$^3$ (reported by Beijing Municipal Environmental Protection Bureau), and the average $A_{\text{Fuchs}}$ measured in Beijing by this campaign was 381.5 μm$^2$/cm$^3$, which is approximately a magnitude higher than those measured in clean environments, such as in Hyytiälä (Dal Maso et al., 2002). Different from comparatively slow

accumulation and depletion process of aerosol concentrations in clean environments, $A_{\text{Fuchs}}$ in Beijing may change rapidly because of changes in air mass origins (Wehner et al., 2008) or accumulation of pollutants.

The sulfuric acid concentration is needed to estimate $L_\Gamma$ and direct measurement of particle size distribution down to ~1 nm will help to better quantify NPF events. Although sulfuric acid has been measured around the world (Erupe et al., 2010) and the analysis based on sub-3 nm size distributions have been conducted sporadically since the development of diethylene glycol scanning mobility particle spectrometer (DEG-SMPS, Jiang et al., 2011a; Jiang et al., 2011b; Kuang et al., 2012) and particle size magnifier (PSM, Vanhanen et al., 2011; Kulmala et al., 2013), there are limited data on

atmospheric sulfuric acid concentrations and directly measured sub-3 nm particle size distributions in China. A campaign in Beijing during 2008 Olympic Games (Yue et al., 2010; Zheng et al., 2011) characterized atmospheric sulfuric acid concentration and its correlation with new particle formation rate. The exponent in the correlation of formation rate, $J_3$, with the sulfuric acid concentration was found to be 2.3. The exponent for correlating derived $J_{1.5}$ with the sulfuric acid concentration was 2.7 (Wang et al., 2011). They were different from the exponents between 1 and 2 often reported in

other places around the world (Riipinen et al., 2007; Sihto et al., 2006; Kuang et al., 2008). The same instrument used in the Beijing campaign was also deployed in Kaiping to measure the sulfuric acid concentration during a one-month campaign in 2008 (Wang et al., 2013a). Sub-3 nm particle size distributions have not been reported previously in China, except for 1-3 nm particle number concentration in Shanghai in Winter 2013 inferred by a PSM (Xiao et al., 2015). Due to the limitation of observation data, although a good correlation between new particle formation rate and the sulfuric

acid concentration in Beijing was found and the ratio of the sulfuric acid concentration over $A_{\text{Fuchs}}$ was reported to positively correlate with number concentration of 3-6 nm particles (Wang et al., 2011), the roles of the sulfuric acid concentration and $A_{\text{Fuchs}}$ in determining the occurrence of NPF events have not been quantitatively illustrated.

In this study, we aimed to examine the roles of $A_{\text{Fuchs}}$ and the sulfuric acid concentration in determining whether a NPF event will occur on a particular day in Beijing. The data analysis was based on simultaneous measurement of particle size

distributions down to ~1 nm and sulfuric acid. The correlation between particle formation rate, $J_{1.5}$, and the sulfuric acid concentration was examined. $L_\Gamma$ was used to predict the occurrence of NPF events. Daily variations of the three parameters determining $L_\Gamma$, i.e., the sulfuric acid concentration, $\Gamma$, and $A_{\text{Fuchs}}$, were compared. A nominal value of $A_{\text{Fuchs}}$ was suggested to predict the occurrence of NPF events in Beijing. The relationship between the PM$_{2.5}$ mass concentration and NPF events was also examined.

**2 Experiments**

A field campaign studying NPF in Beijing was carried out from 7 March 2016 to 7 April 2016. The campaign site was located on the campus of Tsinghua University. Details of this site can be found elsewhere (Cai & Jiang, 2017; He et al.,

2001). A home-made DEG SMPS was used to measure sub-5 nm particle size distributions and a particle size distribution system (including a TSI aerodynamic particle sizer and two parallel SMPSs, equipped with a TSI nanoDMA and a TSI long DMA, respectively) was used to measure size distributions of particles from 3 nm (in electrical mobility diameter) to 10 μm (in aerodynamic diameter, Liu et al., 2016). A specially designed miniature cylindrical differential mobility analyzer (mini- cyDMA) for effective classification of sub-3 nm aerosol was equipped with the DEG-SMPS (Cai et al., 2017). A cyclone was used at the sampling inlet to remove particles larger than 10 μm. The sampled aerosol was subsequently dried by a silica-gel diffusion drier. The diameter change due to drying was neglected when calculating $A_{Fuchs}$ since the mean daytime relative humidity during the campaign period was ~25%. Diffusion losses, charging efficiency, penetration efficiencies through the DMAs, detection efficiencies of particle counters, and multi-charging effect were considered during data inversion. The particle density was assumed to be 1.6 g/cm$^3$ according to local observation results (Hu et al., 2012).

Sulfuric acid was measured by a modified high-resolution time-of-flight chemical ionization mass spectrometer (HR-ToF-CIMS, Aerodyne Research Inc.). Instead of using radioactive ion source, a home-made corona discharge (CD) ion source was utilized with the HR-TOF-CIMS. The CD ion source was designed to be able to operate from a few Torr up to near atmospheric pressure and has been successfully implemented in measuring ambient amine (Zheng et al., 2015a) and formaldehyde (Ma et al., 2016). In this study, nitrate reagent ions were used to measure gaseous sulfuric acid (Zheng et al., 2010). The detailed ion chemistry to generate nitrate ions and the calibration procedure for sulfuric acid measurement have been reported in Zheng et al. (2015b). Ambient sulfuric acid concentration in Beijing has been reported only once in a field campaign conducted in 2008 (Zheng et al., 2011; Wang et al., 2011). Compared to that campaign, the sulfuric acid concentration measured in this study displayed similar diurnal variations, but with lower daily-maximum values. This might be caused by the relatively weak solar radiation intensity encountered in this springtime observation compared with the previous summertime campaign. To verify the precision of sulfuric acid measurement, the instrument was calibrated daily at night and background checks were performed for ~3 minutes each hour during daytime.

A meteorological station (Davis 6250) measuring temperature, relative humidity, wind speed, wind direction, and precipitation was located at ~10 m away from the sampling inlet. The PM$_{2.5}$ mass concentration measured in the nearest national monitoring station (Wanliu station, ~5 km away on the southwest of our campaign site) was also used for analysis. Backward trajectories were obtained from online HYSPLIT server of national oceanic and atmospheric administration (NOAA).

## 3 Theory

Nucleation is only the first step of new particle formation. Gaseous precursors form clusters by random collisions and bound together by Van der Waals force and/or chemical bond. These clusters become particles if they are more likely to grow by condensation rather than evaporate. However, particles formed by nucleation may be scavenged through coagulation with larger particles before they grow large enough to be detected (McMurry, 1983; Zhang et al., 2012). Nucleation only refers to the process that stable molecular clusters formed spontaneously from gaseous precursors. New particle formation also requires subsequent condensational growth of freshly nucleated particles. That is, the occurrence of nucleation is mainly determined by gaseous precursors (e.g., sulfuric acid and organics) in atmospheric environment while new particle formation is also influenced by the coagulation scavenging effect of pre-existing aerosols. A possibility exists that nucleation occurs while NPF events are not observed because of the short lifetime of nucleated particles due to a strong coagulation scavenging (Kerminen et al., 2001). In fact, nucleation can also be suppressed when the aerosol concentration is high since vapours and clusters may also be scavenged by aerosol surface.

Aerosol Fuchs surface area, $A_{Fuchs}$, is a representative parameter of coagulation scavenging based on kinetic theory. It is corrected for particles whose size falls in the transition regime (Davis et al., 1980; McMurry, 1983). The formula assuming unity mass accommodation coefficient (sticking probability) is shown in Eq. (1),

$$A_{Fuchs} = \frac{4\pi}{3} \int_{d_{min}}^{\infty} d_p^{~2} \cdot \left( \frac{Kn + Kn^2}{1 + 1.71Kn + 1.33Kn^2} \right) \cdot n \cdot \mathrm{d}d_p ~, \tag{1}$$

where $d_p$ is particle diameter, $d_{min}$ is the smallest particle diameter in theory and the smallest detected one in practice, $Kn$ is Knudsen number and $n$ is particle size distribution function, $\mathrm{d}N/\mathrm{d}d_p$. The condensation sink and coagulation sink can also describe how rapidly gaseous precursors and particles are scavenged by pre-existing aerosols, respectively (Kerminen et al., 2001; Kulmala et al., 2001). Since the condensation sink is proportional to $A_{Fuchs}$ (McMurry et al., 2005) and the coagulation sink can be approximately converted to the condensation sink using a simple formula (Lehtinen et al., 2007), only $A_{Fuchs}$ is used in this study to describe the coagulation scavenging effect. Condensation sink values reported in previous studies are referred in the form of $A_{Fuchs}$. The diffusion coefficient of sulfuric acid was assumed to be 0.117 cm$^{-2}$s$^{-1}$ (Gong et al., 2010) when converting the condensation sink into $A_{Fuchs}$.

A dimensionless criterion, $L_{\Gamma}$, was proposed to predict the occurrence of NPF events (Kuang et al., 2010). It is defined as,

$$L_{\Gamma} = \frac{\bar{c} \cdot A_{Fuchs}}{4\beta_{11}N_1} \cdot \frac{1}{\Gamma} ~, \tag{2}$$

where $\bar{c}$ is the mean thermal speed of sulfuric acid that can be calculated from molecular kinetic theory; $\beta_{11}$ is the coagulation coefficient between sulfuric acid monomers that can be calculated using Eq. 13.56 in Seinfeld & Pandis (2006); $N_1$ is the number concentration of sulfuric acid; $\Gamma$ is a growth enhancement factor and is defined as,

$$\Gamma = \frac{2GR}{v_1 N_m \bar{c}},\qquad(3)$$

where $GR$ is the observed mean growth rate; $v_1$ is the corresponding volume of sulfuric acid monomer and was estimated to be $1.7\times10^{-28}$ m$^3$ (the volume of a hydrated sulfuric acid molecule, Kuang et al., 2010); $N_m$ is the maximum number sulfuric acid concentration during a whole NPF event period. Since other gaseous precursors in addition to sulfuric acid might also contribute to the condensational growth of particles formed by nucleation (O'Dowd et al., 2002; Ristovski et al., 2010) and only sulfuric acid concentration is used in Eq. (2), the ratio of measured growth rate over the sulfuric acid condensational growth rate (Weber et al., 1997), i.e., $\Gamma$, was used for correction. It should be clarified that $L_\Gamma$ in Eq. (2) is defined similar to that in McMurry et al (2005) but slightly different from that in Kuang et al (2010), since $L_\Gamma$ in this study present time-resolved values rather than event specific ones. Theoretically, $\Gamma$ can also be time and size-resolved if using time and size-resolved GR and time-resolved sulfuric acid (Kuang et al., 2012). However, $\Gamma$ during each NPF event is assumed to be constant in Eq. (3) because further evaluations are needed for this time and size-resolved model. Note that in Eq.(2) the absolute sulfuric acid concentrations were effectively normalized by the corresponding daily-maximum sulfuric acid concentrations and thus has no influence on $L_\Gamma$ values and conclusions based on $L_\Gamma$ reported in this study.

A new balance formula to estimate new particle formation rate was proposed recently (Cai & Jiang, 2017) and is given below,

$$J_k = \frac{\mathrm{d}N_{[d_k,d_u]}}{\mathrm{d}t} + \sum_{d_g=d_k}^{d_{u-1}}\sum_{d_i=d_{\min}}^{+\infty}\beta_{(i,g)}N_{[d_i,d_{i+1}]}N_{[d_g,d_{g+1}]} - \frac{1}{2}\sum_{d_g=d_k}^{d_{u-1}}\sum_{\substack{d_i^3+d_{j+1}^3=d_g^3 \\ d_{i+1}^3+d_j^3=d_g^3 \\ d_i,d_j\geq d_{\min}}}\beta_{(i,j)}N_{[d_i,d_{i+1}]}N_{[d_j,d_{j+1}]} + n_u\cdot GR_u,\qquad(4)$$

where $J_k$ is the formation rate of particles at the size of $d_k$, $N_{[d_k,d_u)}$ is the total number concentration of particles from $d_k$ to $d_u$ (not included), $d_u$ is the upper bound of the size range for calculation (25 nm in this study), $d_{\min}$ is the size of the smallest cluster in theory and the smallest detected size in practice (1.3 nm in this study). The second and third terms in the right hand side of Eq. (4) are the coagulation sink term (*CoagSnk*) and the coagulation source term (*CoagSrc*), respectively. The difference between *CoagSnk* and *CoagSrc* is the net *CoagSnk* representing the net rate of particles from $d_k$ to $d_u$, i.e., lost by coagulation scavenging. The last term is often negligible according to the determination criterions for $d_u$. d$N$/d$t$ is the balance result of $J_k$ and net *CoagSnk*.

## 4 Results and Discussion

A total of 26 days from 12 March to 6 April was classified by the occurrence of daytime NPF event. A typical NPF day is featured with distinct and persisting increases in the sub-3 nm particle number concentration and subsequent growth of these nucleated particles. A non-event day means that neither of these two features was observed. As shown in Fig. 1, there are 11 typical NPF days and 13 non-event days. The rest 2 days, i.e., Mar. 19[th] and Mar. 30[th], were classified as undefined days. On these days, the increase in the sub-3 nm particle number concentration and subsequent growth were

both observed. However, the sub-3 nm particle number concentration was relatively low and the evolution of particle size distributions was not continuous. NPF events mainly occurred when wind came from northwest of Beijing and non-event days were associated with air masses from southwest (as summarized in Table 1). Air masses coming from north usually experience less influence from urban pollution (Wehner et al., 2008; Wang et al., 2013b), i.e., the $A_{Fuchs}$ values on days dominated by the north wind are usually lower than those on days dominated by the southwest wind (Wu et al., 2007).

The occurrence of NPF events in most days can be predicted by $L_{\Gamma}$ if unity was empirically chosen as the threshold value. Greater $L_{\Gamma}$ indicates higher possibilities of nucleated particles to be scavenged by coagulation before they can continue to grow. Growth rates on non-event days were assumed to be 2.4 nm/h, the mean value of observed growth rates on NPF days (the range is 1.2 nm/h to 3.3 nm/h). A threshold value of $L_{\Gamma}$ can not be theoretically predicted but can be empirically estimated. 0.7 was suggested as the threshold value by Kuang et al. (2010). However, unity suggested by McMurry et al.

(2005) appeared to work better for results from this campaign in Beijing. As shown in Table 1, the median and mean values of $L_{\Gamma}$ on NPF days observed in this campaign were 0.55 and 0.71 (with a standard deviation of 0.40), respectively, comparing to 3.05 and 3.45 on non-event days (with a standard deviation of 1.79), respectively. However, some exceptions were also observed. On the two undefined days, $L_{\Gamma}$ were 1.40 and 0.64, respectively, and weak nucleation was observed. Although the estimated $L_{\Gamma}$ value on 18[th] March was 1.75, a comparatively weak but still distinct NPF event was

observed. Despite these few exceptions, $L_{\Gamma}$ works well in most days in this campaign and were verified in other places (Kuang et al., 2010). The following discussion is focused on the contribution of different factors, i.e., the sulfuric acid concentration, $\Gamma$, and $A_{Fuchs.}$

### 4.1 The Role of Gaseous Precursors

There was a positive correlation between the estimated new particle formation rate, $J_{1.5}$, and the sulfuric acid concentration

during most NPF periods (typically 8:00-16:00 when the estimated $J_{1.5}$ was greater than zero). On NPF days, an increase in the sub-3 nm particle number concentration was often accompanied with an increase in the sulfuric acid concentration (as shown in Fig. 2). Considering the possible sensitivity of the fitted parameters to the fitting time period (Kuang et al., 2008), the correlation between $J_{1.5}$ and the sulfuric acid concentration was only examined for NPF periods. We found that the mean coefficient of determination ($R^2$) in this campaign was 0.53. The exponents for correlating the $J_{1.5}$ and the

sulfuric acid concentration ranged from 1.5 to 4.0 in the 10 days with a mean value of 2.4 (29[th] March was not included

because of insignificant correlation). This is in consensus with previously reported mean exponent of 2.3 using $J_3$ in

Beijing (Wang et al., 2011). However, the exponent is quite different from the exponents no greater than 2 observed in

North America and Europe (Kuang et al., 2008; Riipinen et al., 2007; Sihto et al., 2006), indicating that activation or

kinetic nucleation alone can not explain all NPF events observed in this campaign.

Although the correlation between the sulfuric acid concentration and the particle formation rate was significant, sulfuric

acid appeared not to be the determining factor for whether a NPF event would occur in Beijing. As illustrated by the

temporal trend of the sulfuric acid concentration in Fig. 2, a significant diurnal variation was observed every day.

However, the differences among the daily-maximum sulfuric acid concentrations were small. The variations of daily-

maximum sulfuric acid concentration were significantly less than those of $A_{Fuchs}$. The geometrical standard deviation and

relative standard deviation of maximum sulfuric acid concentration on each day were 1.40 and 0.34, respectively, while

those of the daily-averaged $A_{Fuchs}$ values were 2.56 and 0.82, respectively. The sulfuric acid concentrations during NPF

periods were not significantly higher than those between 8:00 - 16:00 on non-event days (significant value, p=1). In

addition, comparatively high sulfuric acid concentrations, e.g., on 4[th] - 6[th] April, did not necessarily lead to NPF events.

The influence of growth enhancement factor, Γ, on the occurrence of NPF events also needs to be addressed because

sulfuric acid alone may not explain the observed growth rates. Estimated Γ value for each event was normalized by the

geometric mean Γ value for the whole campaign to make it comparable with those obtained from previous studies (Kuang

et al., 2010): MILAGRO in Tecamac (Iida et al., 2008); ANARChE (McMurry et al., 2005) in Atlanta; Boulder (Iida et

al., 2006); QUEST II (Sihto et al., 2006), QUEST IV (Riipinen, et al., 2007), and EUCAARI (Manninen et al., 2009) at

the SMEAR II station in Hyytiälä. It should be clarified that the relative value of Γ can improve the comparability by

overcoming some uncertainties in the measured sulfuric acid concentrations in different studies. Fig. 4 indicates that Γ

values observed in this study distribute in a relatively narrow range, similar to those observed in Tecamac, Atlanta, and

Boulder, while different from the wide-spreading characteristics of Γ values in Hyytiälä. Geometric standard deviations

of Γ values were 1.31, 1.75, 2.23, 1.87, 1.62, 2.77, and 2.87 in this campaign, MILAGRO, ANARChE, Boulder, QUEST

II, QUEST IV, and EUCAARI, respectively. The daily variations of Γ values in Beijing were less than those observed in

other places. They were also less than the daily variations of $A_{Fuchs}$ values measured in this campaign. Considering the

small daily variations of both the sulfuric acid concentration and Γ values, it is reasonable to conclude that the abundance

of gaseous precursors, such as sulfuric acid, in Beijing during the campaign period was sufficiently high for nucleation

to occur but the occurrence of NPF events appeared to be governed by $A_{Fuchs}$.

## 4.2 Relationship between $A_{Fuchs}$ and NPF Events

Comparatively lower $A_{Fuchs}$ values were found during most of the NPF days whereas the sulfuric acid concentrations on NPF days were not significantly higher than those on non-event days. NPF events mainly occurred when $A_{Fuchs}$ was smaller than 200 $\mu m^2/cm^3$ (the corresponding condensation sink is 0.027 $s^{-1}$). Non-event days mainly corresponded to a real-time $A_{Fuchs}$ value greater than 200 $\mu m^2/cm^3$ and an average $A_{Fuchs}$ value greater than 350 $\mu m^2/cm^3$ (Fig. 5). The value of 200 $\mu m^2/cm^3$ appeared to be an empirical division between NPF days and non-event days. If $A_{Fuchs}$ was lower than this value, a NPF event tended to occur. Otherwise, the occurrence of NPF events was suppressed because of the predominant coagulation scavenging effect. A similar threshold (the condensation sink of 0.02 $s^{-1}$) was found in Budapest, Hungary (Salma et al., 2017).

The variation of $L_\Gamma$ in Beijing was governed by $A_{Fuchs}$. The measured $L_\Gamma$ and $A_{Fuchs}$ values were in a good correlation with coefficient of determination ($R^2$) of 0.88. The mean relative error of fitted $L_\Gamma$ using $A_{Fuchs}$ was 11.4% compared to the measured ones (Fig. 6(a)). It should be clarified that $GR$ on non-event days in this campaign was assumed to be the same (2.4 nm/h, an average of the fitted values on NPF days). The correlation between $L_\Gamma$ and $A_{Fuchs}$ on NPF days alone had $R^2$ of 0.89. The $A_{Fuchs}$ of 200 $\mu m^2/cm^3$ corresponds to an $L_\Gamma$ of approximate unity in this campaign. Since $L_\Gamma$ has been verified as a proper nucleation criterion in diverse atmospheric environments, it is reasonable to conclude that $A_{Fuchs}$ was the governing factor of the occurrence of NPF events observed in this campaign.

The characteristics of $A_{Fuchs}$ dominated NPF events in Beijing are different from those at other locations. As shown in Fig. 6(b), $L_\Gamma$ and $A_{Fuchs}$ in most other places do not correlate well, indicating that $A_{Fuchs}$ alone can not predict the occurrence of NPF events at these locations. The variations of these parameters at various locations are illustrated in Fig. 7. In Atlanta and Boulder, $A_{Fuchs}$ values fluctuated within relatively narrow ranges while the concentrations of gaseous precursors participating in nucleation differed significantly. The variations of $L_\Gamma$ at these locations were mainly caused by the relatively large variations in the concentrations of gaseous precursors. However, the contribution of gaseous precursors to $L_\Gamma$ in Beijing was relatively stable and the variations of $L_\Gamma$ were mainly caused by the variations in $A_{Fuchs}$ values.

The predominant role of $A_{Fuchs}$ in Beijing can also be explained using the balance formula shown as Eq. (4). It is d$N$/d$t$ rather than the formation rate, $J$, that directly reflects whether a NPF event has occurred or not. d$N$/d$t$ is the balanced result of the formation rate and the net $CoagSnk$. Different from $L_\Gamma$ that is the ratio of the particle loss rate over the growth rate, the ratio of the net $CoagSnk$ over $J$ represents how many nucleated particles are lost due to the coagulation scavenging. The surviving particles are accounted for by the increment in number concentration of particles in the nucleation mode (1-25 nm). The nucleation mode was used in this study to estimate d$N$/d$t$ caused by nucleation because newly formed particles seldom grew beyond 25 nm in the evaluated time period. Surviving possibilities of nucleated particles can also be inferred using the growth rate and $A_{Fuchs}$ (Weber et al., 1997; Kerminen & Kulmala, 2002; Kuang et

al., 2012). However, the ratio of the net *CoagSnk* over *J* was used because it is based on measured particle size distributions. Note that theoretically the ratio of the net *CoagSnk* over *J* can be greater than unity. This would correspond to a negative $dN/dt$ value. For better description of the occurrence of NPF events rather than the whole process including termination, only NPF periods when $dN/dt$ was positive were considered here. On average, 70% of particles formed by nucleation were lost due to coagulation scavenging on NPF days (as shown in Fig. 8), indicating high coagulation losses in Beijing even on NPF days. When the $A_{Fuchs}$ value was much greater, most nucleated particles were lost due to the coagulation scavenging rather than were grown into to larger sizes, such that NPF events were less likely to be observed. It should be clarified that although with much less possibility, NPF events may also occur in Beijing when $A_{Fuchs}$ was greater than 200 $\mu m^2/cm^3$. In this campaign, a distinct NPF event was observed with a comparatively high $A_{Fuchs}$ value of 329 $\mu m^2/cm^3$ (on 18[th] March). It was significantly higher than the suggested threshold value of 200 $\mu m^2/cm^3$. As indicated by Table 1, this exception was caused by the failure of $L_\Gamma$ rather than $A_{Fuchs}$ alone, i.e., NPF events occurred when estimated $L_\Gamma$ was greater than unity (the empirical threshold value). The comparatively low number concentration of sub-3 nm particles together with the moderate particle formation rate indicated that the NPF event was suppressed. In addition, previous studies in Beijing also observed some NPF events when $A_{Fuchs}$ values were relatively high (Wu et al., 2007; Wang et al., 2013c; Wang et al., 2017), e.g., an $A_{Fuchs}$ value of ~555 $\mu m^2/cm^3$ (Kulmala et al., 2016). These reported $A_{Fuchs}$ values might be overestimated since the daily-average value rather than the average only over NPF event periods was used. $A_{Fuchs}$ in Beijing during non-event periods can be significantly higher. Nevertheless, $A_{Fuchs}$ can be considered as the major determining factor of the occurrence of NPF events in Beijing while admitting that exceptions can occasionally occur at a medium $L_\Gamma$ value greater than unity (corresponding to the $A_{Fuchs}$ value of 200 $\mu m^2/cm^3$).

**4.3 A case Study of 3 Days**

Three continuous days including two NPF days and one non-event day are shown in Fig. 9 to further illustrate the roles of $A_{Fuchs}$ and sulfuric acid (together with other gaseous precursors) in affecting the occurrence of NPF events in Beijing. On 2[nd] April, $A_{Fuchs}$ remained at a relatively low level. A NPF event occurred after sunrise (together with an increase in the sulfuric acid concentration) and ended in the afternoon when the sulfuric acid concentration decreased to a low level. The whole NPF event began at approximately 7:30 and ended at approximately 14:30 that was also the typical time period for other NPF events observed in this campaign. However, when wind direction changed from northwest to southwest at the noon of on 3[rd] April, the sulfuric acid concentration decreased and $A_{Fuchs}$ increased rapidly because of particles transported from south. This led to an increase in $L_\Gamma$. The ongoing NPF event was interrupted and no newly nucleated particles were detected even when the sulfuric acid concentration increased again later. On 4[th] April, $A_{Fuchs}$ stayed at a high level. $L_\Gamma$ was always greater than unity. The maximum sulfuric acid concentrations on 4[th] April were even higher than those on 2[nd] and 3[rd] April. However, no NPF event was observed. It supports the argument that the abundance of

gaseous precursors in Beijing are often high enough for nucleation to happen, however, whether or not a NPF event occurs is mainly governed by $A_{Fuchs}$.

**4.4 Predicting NPF Days Using PM$_{2.5}$ Mass Concentration**

The PM$_{2.5}$ mass concentration in Beijing serves as a rough but simple parameter to predict whether a NPF event can
happen. The value of $A_{Fuchs}$ is affected by particle size distributions. Accumulation mode particles ranging from 50 nm to 500 nm in Beijing were the major contribution to $A_{Fuchs}$. Normalized size distributions of accumulation mode particles were relative stable at various $A_{Fuchs}$ levels (as shown in Fig. 10). On NPF days when $A_{Fuchs}$ were relatively low, particles smaller than 30 nm in diameter formed by nucleation and subsequent growth also contributed to $A_{Fuchs}$, although $A_{Fuchs}$ was still governed by accumulation mode particles. Thus, $A_{Fuchs}$ should show better correlation with the particle mass
concentration rather than the particle number concentration. Figure 11 indicates that there was a good correlation between $A_{Fuchs}$ and the PM$_{2.5}$ mass concentration in Beijing with $R^2$ of 0.85, although the correlation at a high $A_{Fuchs}$ level was generally better than that at a low $A_{Fuchs}$ level because particles formed by nucleation significantly changed the shape of particle size distribution functions on NPF days. Measured PM$_{2.5}$ mass concentrations in the 26 days ranged from 3 to 420 μg/m$^3$, wide enough to represent both relative clean days and severely polluted days in Beijing. The PM$_{2.5}$ mass
concentrations during NPF event periods were mostly lower than 30 μg/m$^3$, except for the event on 18$^{th}$ March. On non-event days, the PM$_{2.5}$ mass concentrations between 8:00 and 16:00 were typically greater than 30 μg/m$^3$. Note that this threshold PM$_{2.5}$ value of 30 μg/m$^3$ may not be valid for the whole year. This campaign was in March and early April. Emissions and radiation intensity are different in different seasons, such that the concentrations of gaseous precursors can vary with seasons as well.

The criterion of PM$_{2.5}$ mass concentration was applied to predict NPF events measured at the same site in Beijing in April and May, 2014. Among 38 days in that campaign, 11 typical NPF events were identified. For 9 NPF events, average PM$_{2.5}$ mass concentrations during event periods were lower than 30 μg/m$^3$. For the other 2 events, it was 49.8 and 40.5 μg/m$^3$, respectively. In another campaign in Beijing during January 2016 (Jayaratne et al., 2017), 14 NPF events were observed. Among them, 12 events occurred when the daily-average PM$_{2.5}$ mass concentration was lower than 30 μg/m$^3$.
The daily-average PM$_{2.5}$ mass concentrations on 16 non-event days were all greater than 40 μg/m$^3$.

**5 Conclusions**

Factors governing the occurrence of NPF events in Beijing were examined using data from a field campaign during 12 March 2016 to 6 April 2016. In these 26 days, 11 typical NPF events were observed. The rest were 2 undefined days and 13 non-event days. The new particle formation rate, $J_{1.5}$, was in positive correlation with the sulfuric acid concentration
with a fitted mean exponent of 2.4. However, the sulfuric acid concentrations on NPF days were not significantly higher

than those on non-event days. A dimensionless criterion proposed by Kuang et al. (2010), $L_\Gamma$, was found to be applicable to predict NPF events in most days. Theoretically, $L_\Gamma$ is determined by the sulfuric acid concentration, the enhancement factor, $\Gamma$, and aerosol Fuchs surface area, $A_{Fuchs}$, together. In Beijing, however, $A_{Fuchs}$ alone was found to be in a good correlation with $L_\Gamma$ ($R^2 = 0.88$). Different from NPF events observed at other locations, such as Hyytiälä, the daily-

maximum sulfuric acid concentration and the enhancement factor in Beijing only varied in a narrow range with geometric standard deviations of 1.40 and 1.31, respectively, while $A_{Fuchs}$ varied significantly among days with a geometric standard deviation of 2.56. It was inferred that the concentrations of gaseous precursors, such as sulfuric acid, in Beijing were high enough to initiate nucleation while it was $A_{Fuchs}$ that determined whether a NPF event would occur or not. An $A_{Fuchs}$ value of 200 $\mu m^2/cm^3$ was proposed as the empirical threshold in Beijing below which NPF events are highly likely to occur.

NPF events will be suppressed when $A_{Fuchs}$ is higher than this threshold value. The $A_{Fuchs}$ dominated characteristics in Beijing are different from those at other locations, such as Atlanta, Boulder, and Hyytiälä. Since $A_{Fuchs}$ in Beijing was mainly governed by accumulation mode particles (50 to 500 nm) and the normalized $dA_{Fuchs}/dlogd_p$ in this size range was relatively stable at different $A_{Fuchs}$ levels in Beijing, measured $A_{Fuchs}$ was in a good correlation with the $PM_{2.5}$ mass concentration ($R^2 = 0.85$). Accordingly, the $PM_{2.5}$ mass concentration may also serve as a rough and simple parameter to

predict the occurrence of NPF events in Beijing. An empirical $PM_{2.5}$ threshold value of 30 $\mu g/m^3$ was proposed based on data from this field campaign and was found to also work well for other field campaigns in Beijing.

**Acknowledgement**

Financial supports from the National Science Foundation of China (21422703, 41227805, 21521064, 21377059 & 41575122) and the National Key R&D Program of China (2014BAC22B00, 2016YFC0200102 & 2016YFC0202402)

are acknowledged.

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

**Table 1: Characteristics of each campaign day.**

| Date (mm/dd) | Classification | Max $J_{1.5}$ ($cm^{-3}s^{-1}$) | $N_{1-3}$ (#/$cm^3$) | $A_{Fuchs}$ ($\mu m^2/cm^3$) | $L_T$ | Wind direction[*] |
|---|---|---|---|---|---|---|
| 03/12 | Non-event | – | 0 | 919.5 | 3.63 | SW |
| 03/13 | NPF | 156.0 | 26347.5 | 119.7 | 0.71 | NW |
| 03/14 | Non-event | – | 0 | 632.7 | 3.05 | NW |
| 03/15 | Non-event | – | 0 | 733.9 | 3.73 | SW |
| 03/16 | Non-event | – | 0 | 796.2 | 4.15 | WSW |
| 03/17 | Non-event | – | 0 | 1140.1 | 9.04 | WSW |
| 03/18 | NPF | 33.8 | 741.2 | 329.0 | 1.75 | WNW |
| 03/19 | Undefined | Weak[**] | 1643.67 | 240.8 | 1.40 | SE |
| 03/20 | Non-event | – | 137.9 | 348.8 | 1.74 | NNW |
| 03/21 | Non-event | – | 0 | 512.0 | 2.76 | SSW |
| 03/22 | Non-event | – | 0 | 457.6 | 2.58 | E |
| 03/23 | NPF | 30.1 | 3846.3 | 76.1 | 0.57 | NNW |
| 03/24 | NPF | 46.8 | 5576.7 | 145.2 | 0.76 | NNW |
| 03/25 | NPF | 57.0 | 4637.7 | 126.7 | 0.52 | NNE |
| 03/26 | NPF | 41.5 | 9640.9 | 100.4 | 0.71 | N |
| 03/27 | NPF | 31.2 | 2806.2 | 90.6 | 0.44 | NW |
| 03/28 | Non-event | – | 0 | 508.1 | 2.86 | W |
| 03/29 | NPF | 32.3 | 2449.8 | 121.0 | 0.69 | NW |
| 03/30 | Undefined | 17.7 | 2885.7 | 88.8 | 0.64 | NW |
| 03/31 | Non-event | – | 0 | 767.0 | 4.21 | SW |
| 04/01 | NPF | 50.9 | 5477 | 51.7 | 0.22 | WNW |
| 04/02 | NPF | 46.9 | 10002 | 63.1 | 0.31 | NW |
| 04/03 | NPF | 21.6 | 10962.9 | 105.7 | 0.24 | NW |
| 04/04 | Non-event | – | 442 | 398.2 | 3.09 | SW |
| 04/05 | Non-event | – | 185 | 391.2 | 2.33 | NW |
| 04/06 | Non-event | – | 0 | 365.5 | 1.71 | SW |

**\*: Indicated by 12-hour backward trajectory (starting at noon, 500 m in altitude).**

**\*\*: Difficult to estimate.**


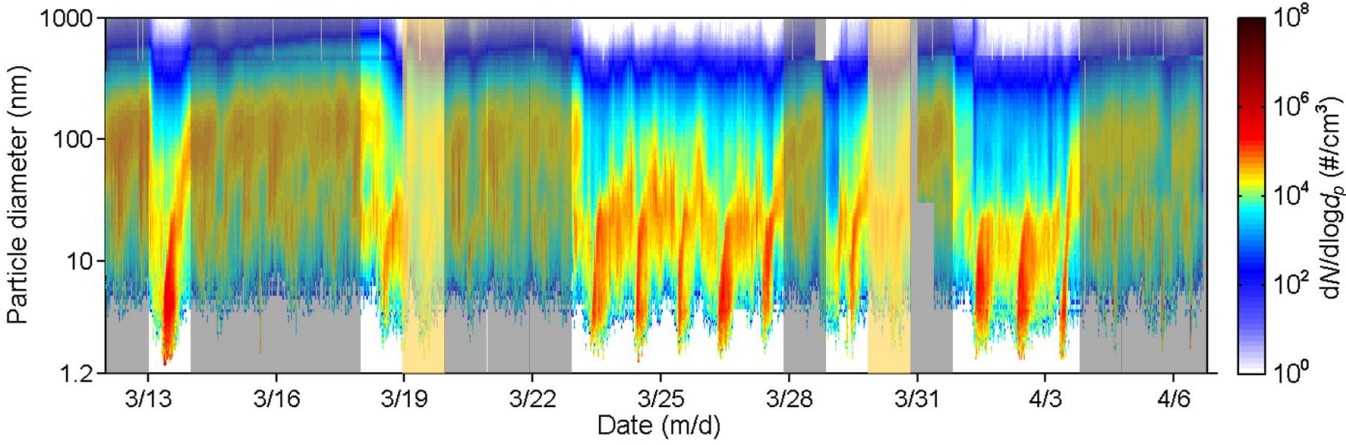

**Figure 1 Contour of measured particle size distributions during 12 March to 6 April. Identified thirteen non-event days and two undefined days are shadowed by grey and yellow background, respectively.**


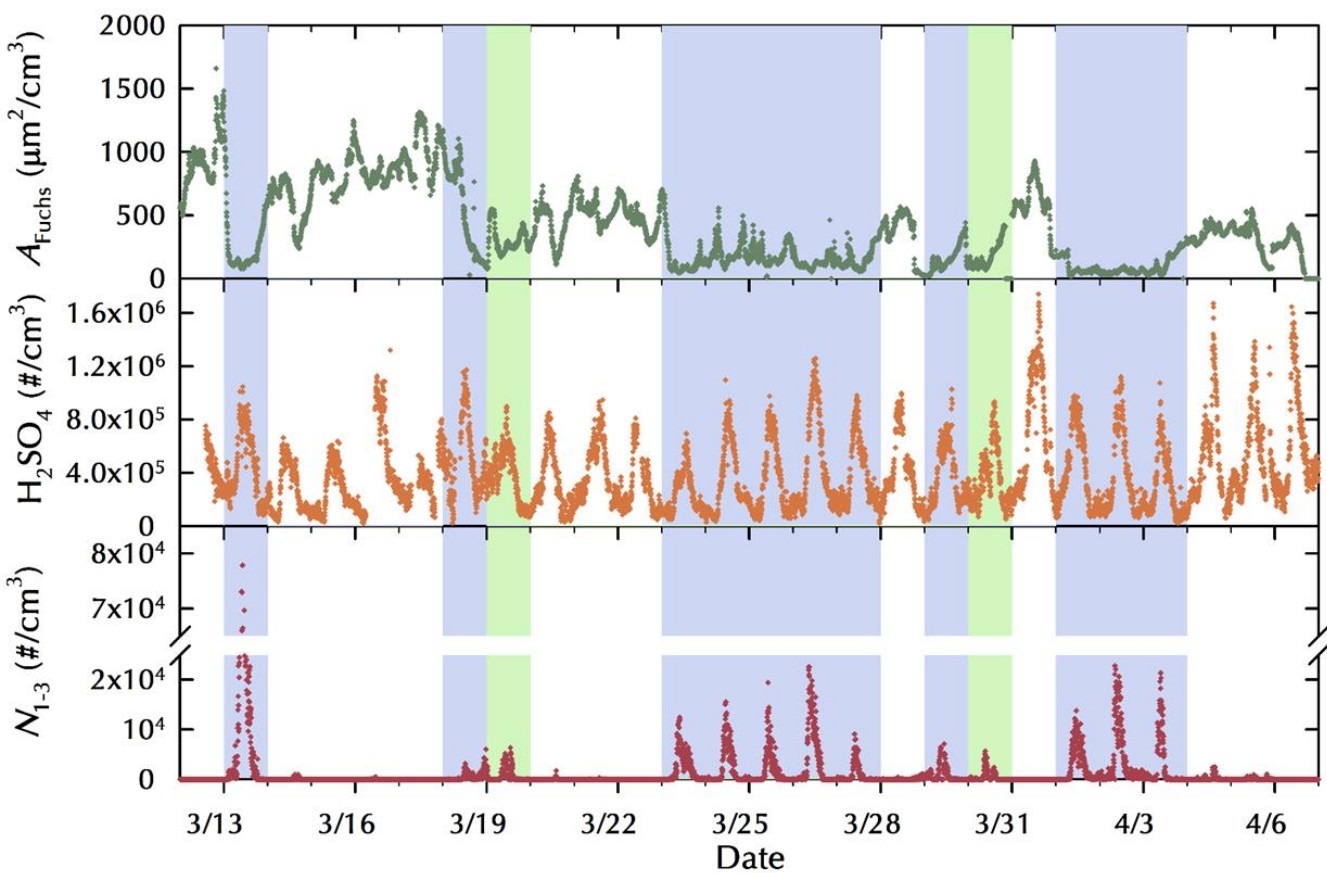

**Figure 2: Time series for Fuchs surface area ($A_{Fuchs}$), the sulfuric acid concentration, and number concentration of 1-3 nm particles. Typical NPF days and undefined days are shadowed by light blue and light green background, respectively.**

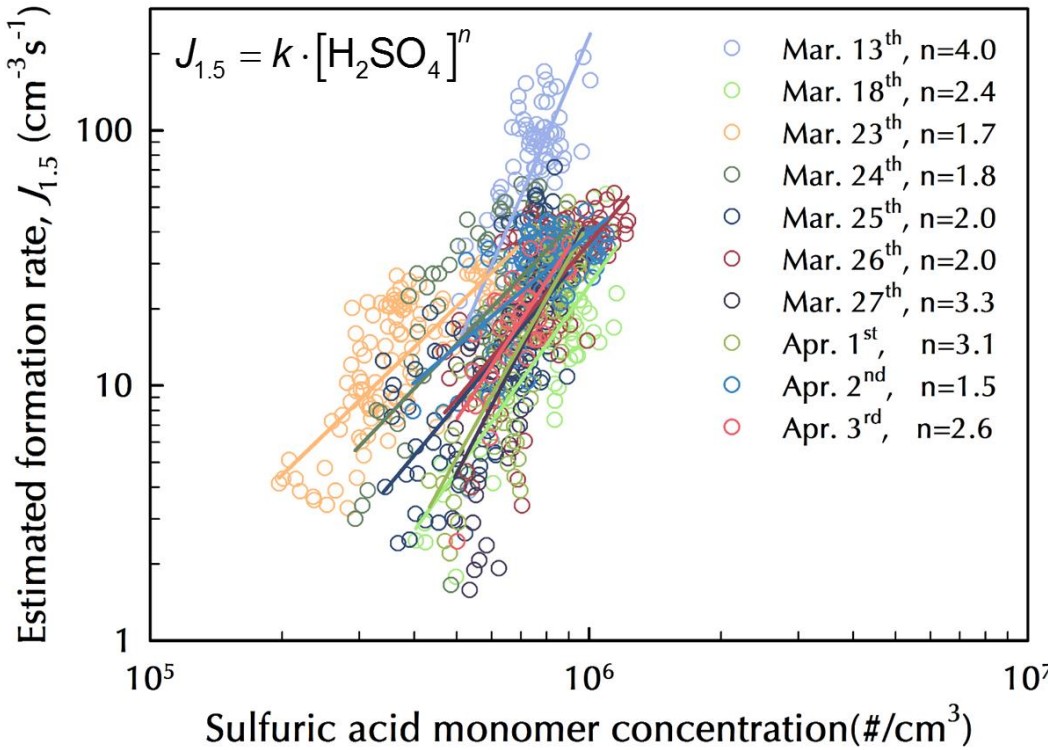


**Figure 3: The correlations between the estimated new particle formation rate, $J_{1.5}$, and the sulfuric acid concentration during**

**NPF event period on each NPF day. The regression line of $J_{1.5}$ versus the sulfuric acid concentration was exponentially fitted.**

**n is the exponent. Data on 29[th] March was not included because the correlation was not significant (p = 0.34).**

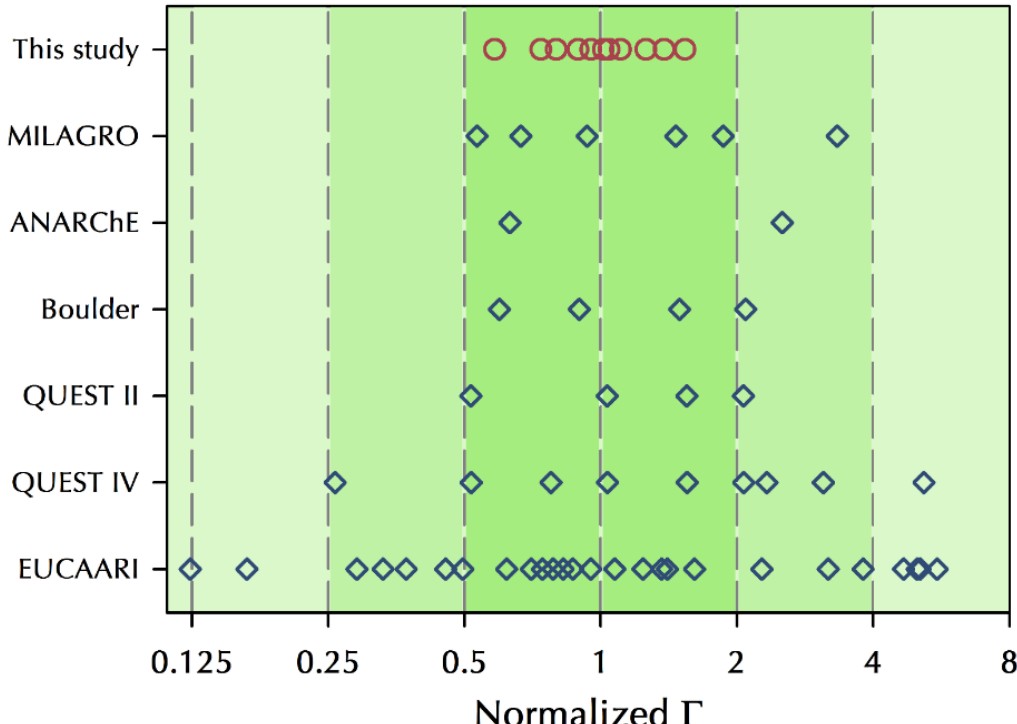


**Figure 4: Normalized growth enhancement factor, Γ, in this campaign in comparison to those reported for other campaigns. Γ was normalized by the geometric mean value in each campaign.**

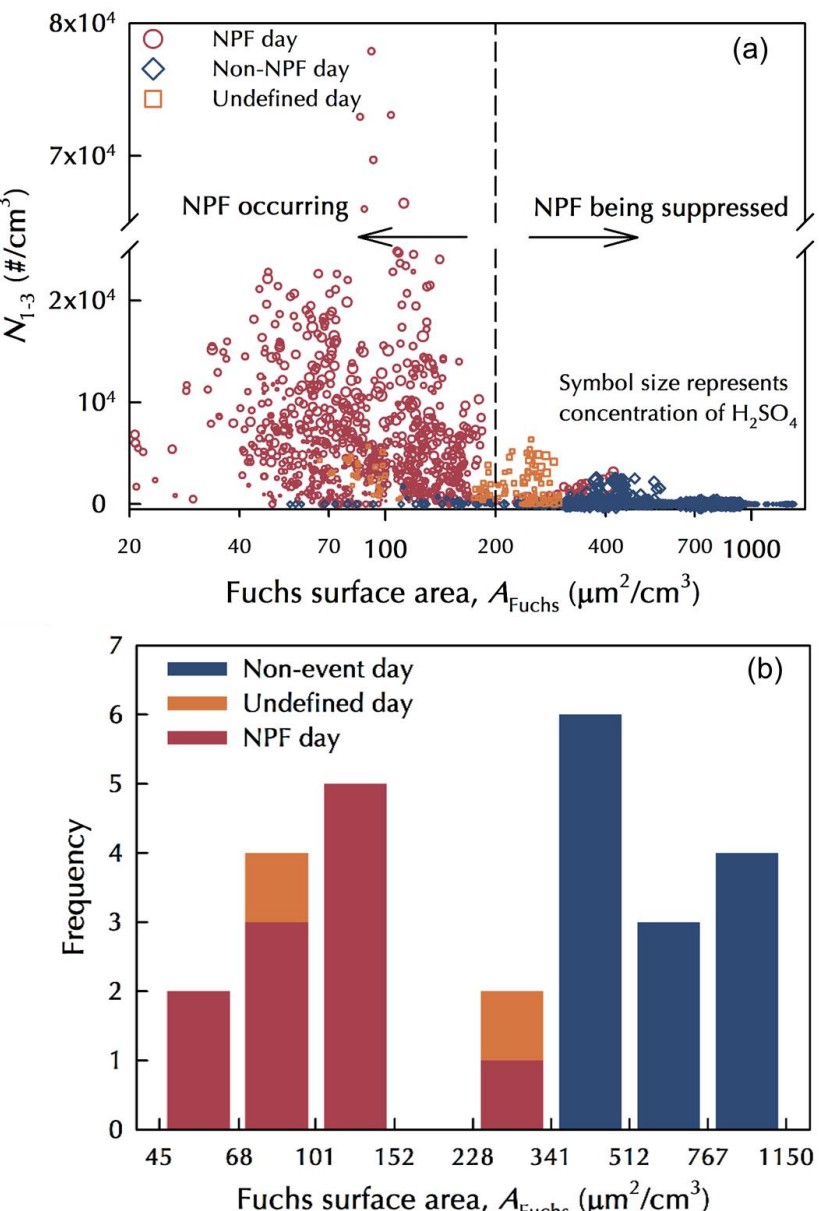

**Figure 5: (a) The relationship between Fuchs surface area and number concentration of 1-3 nm particles, $N_{1-3}$. The relative concentration of measured sulfuric acid is represented by symbol size, i.e., the higher the relative concentration, the bigger the symbol size. Data points are 5-minute-resolved. (b) Frequencies of observed NPF days, undefined days and non-event days in comparison to the daily-average $A_{Fuchs}$. On typical NPF days and undefined days, $A_{Fuchs}$ was averaged during NPF event periods. On non-event days, it was averaged between 8:00 and 16:00. $A_{Fuchs}$ values were binned in logarithmic scale ranging from 45 to 1150.**

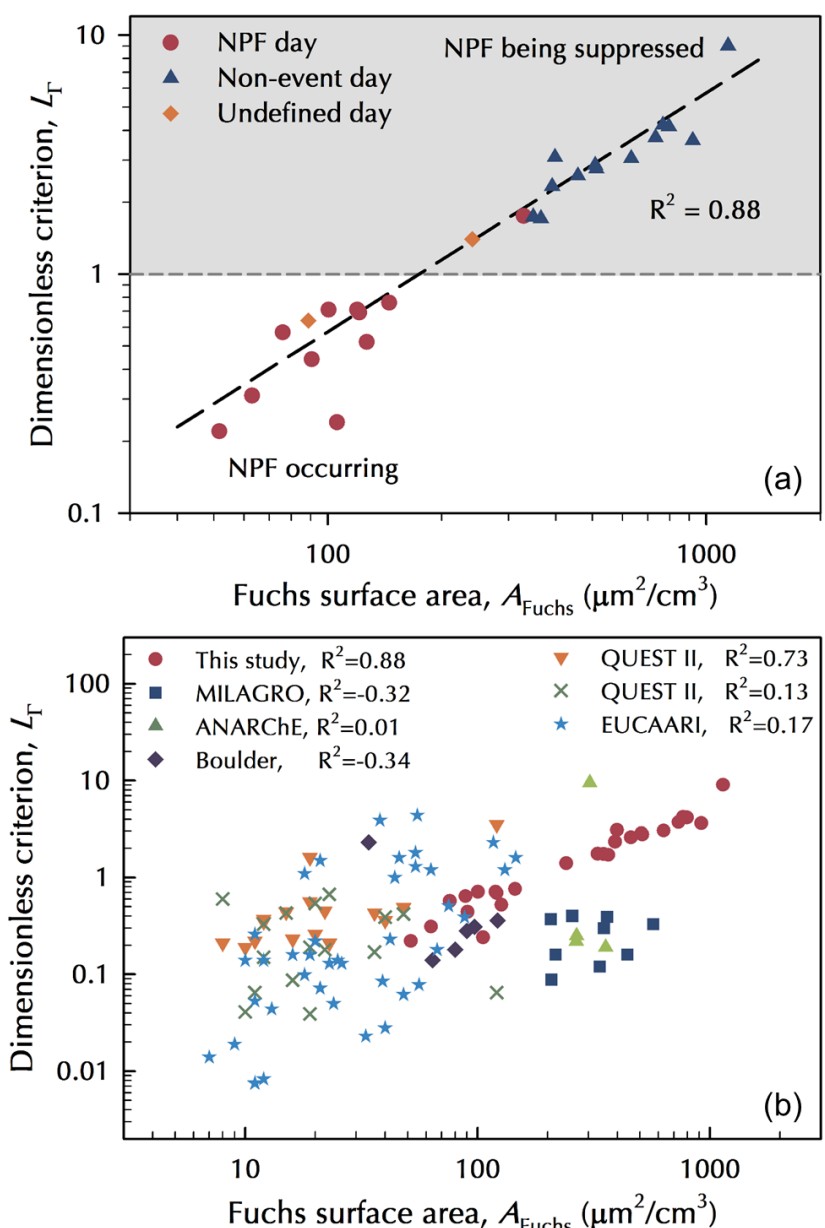

**Figure 6: (a) The correlation between $L_\Gamma$ and $A_{Fuchs}$ (data from Table 1) in this campaign. NPF days, non-event days, and undefined days are shown as different symbols. The regression was based on all campaign days. (b) The correlation between**

**$L_\Gamma$ and $A_{Fuchs}$ estimated for this study in comparison to other campaigns.**

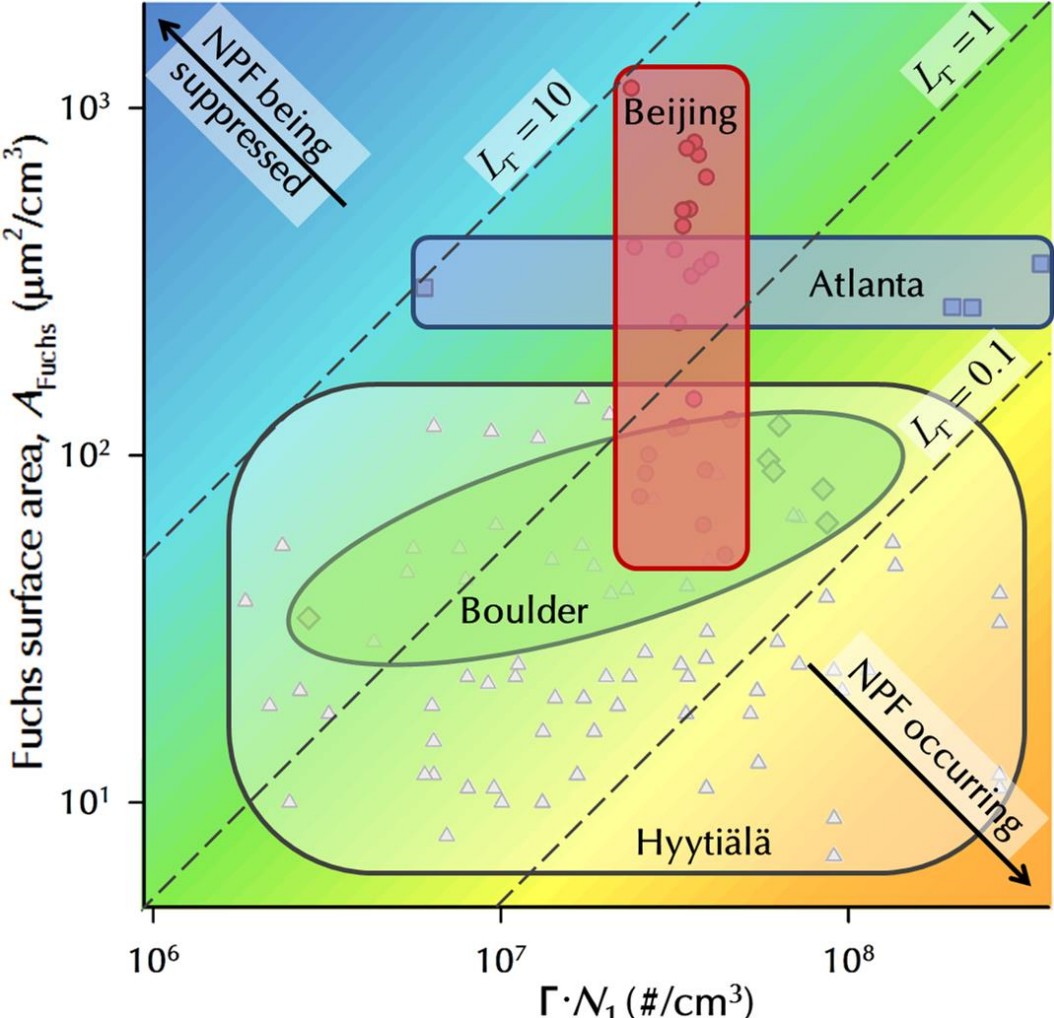

**Figure 7: The schematic of governing factors for $L_\Gamma$ at different locations. Concentration of growth relevant gaseous precursors is represented by $\Gamma \cdot N_1$, where $\Gamma$ is the growth enhancement factor and $N_1$ is the sulfuric acid number concentration.**
**Background colour represents the magnitude of $L_\Gamma$. Data for each location are shown as different symbols (circle: Beijing; square: Atlanta; diamond: Boulder; triangle: Hyytiälä). The ellipse and the boxes were artificially drawn to illustrate the variations. Tecamac was not included due to the lack of data on non-event days. Both axes are in log scale.**

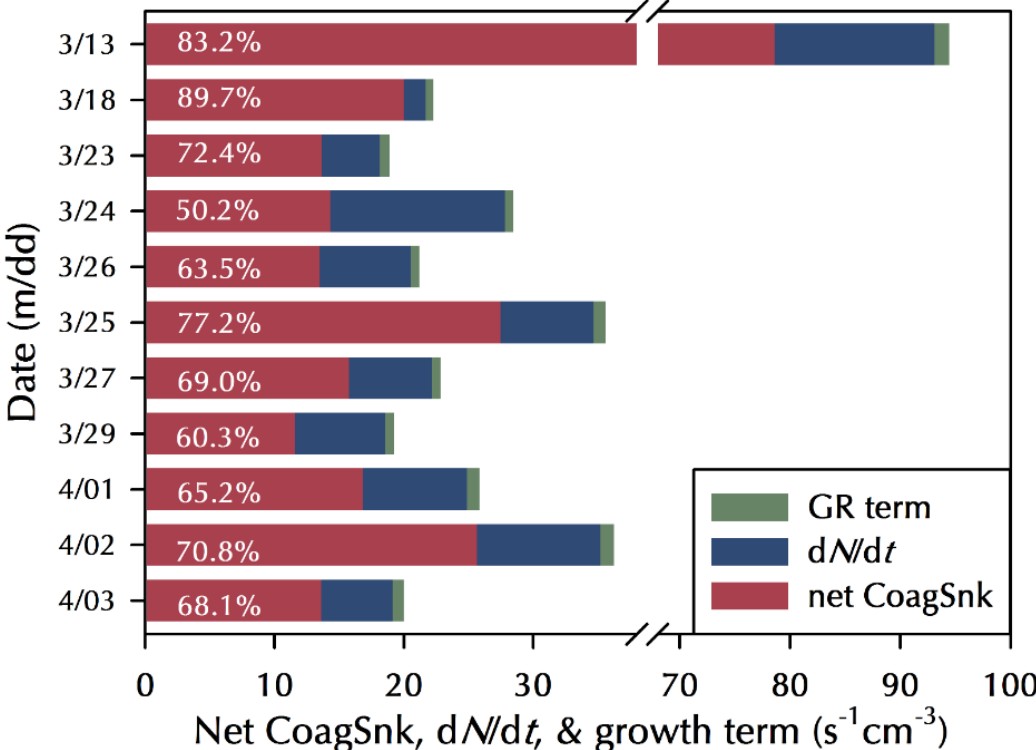

Figure 8: Average contribution of the net CoagSnk, d$N$/d$t$, and the condensational growth term (GR term) to the estimated new particle formation rate, $J_{1.5}$, on identified typical NPF days. The percentage presented in each column is the relative ratio of the net CoagSnk compared to $J_{1.5}$ of that NPF event. Note that only the time period when d$N$/d$t$ was positive during a NPF event was taken in to account when calculating average contribution.

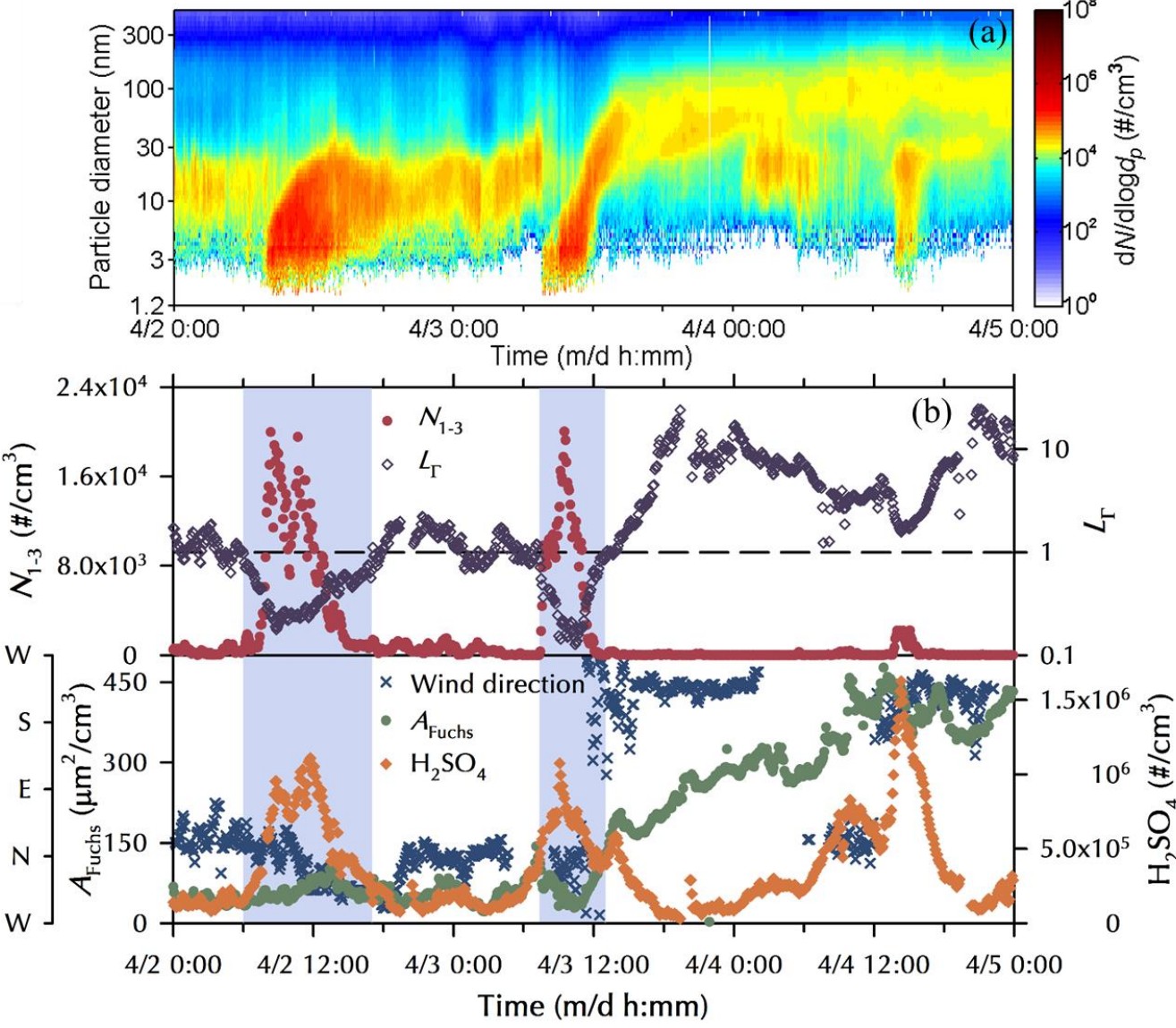


**Figure 9: (a) Contour of measured particle size distributions on 2nd, 3rd, and 4th April. (b) Representative parameters on these three NPF days. Time periods when $L_\Gamma$ was lower than 1.0 are shadowed by light blue background. When wind speed was close to zero, the corresponding wind direction data were not included in the plot.**

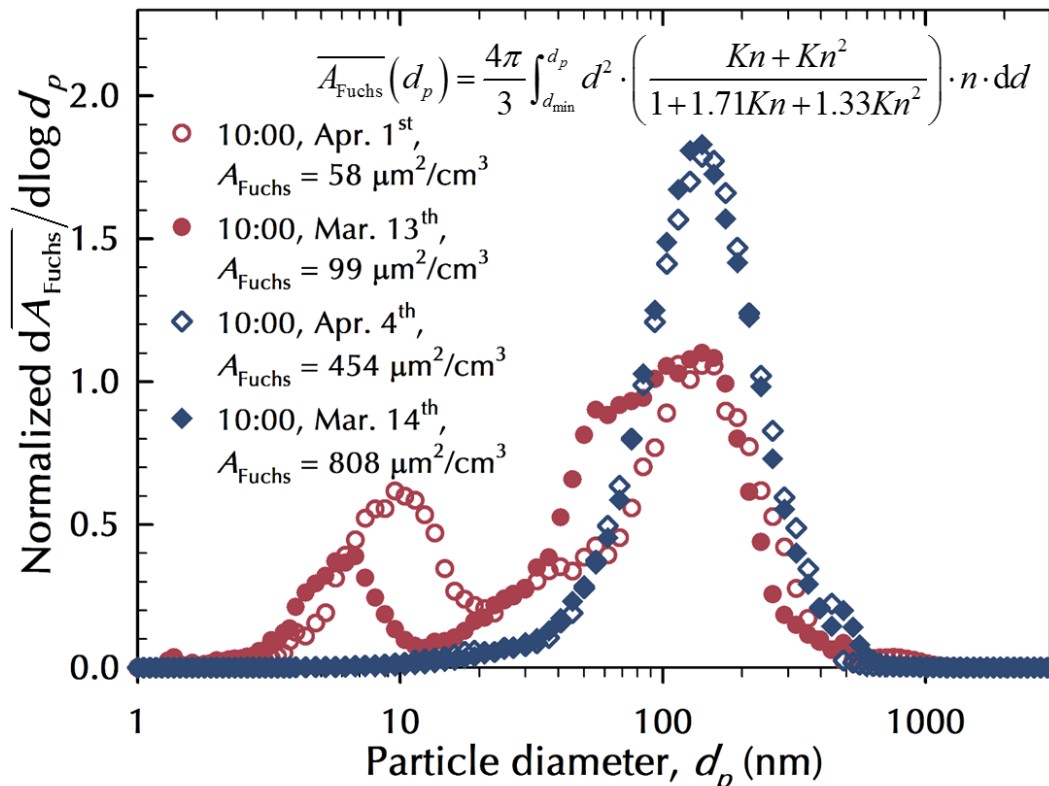


**Figure 10: Normalized distribution of cumulative Fuchs surface area,** $\overline{A_{\text{Fuchs}}}$ **, as a function of the particle diameter,** $d_p$**, on two NPF days (red circle) and two non-event days (blue diamond).** $\overline{A_{\text{Fuchs}}}$ **is equal to** $A_{\text{Fuchs}}$ **when** $d_p$ **is approaching positive infinity.** $\text{d}\overline{A_{\text{Fuchs}}}/\text{d}\log d_p$ **is normalized by** $A_{\text{Fuchs}}$**.**

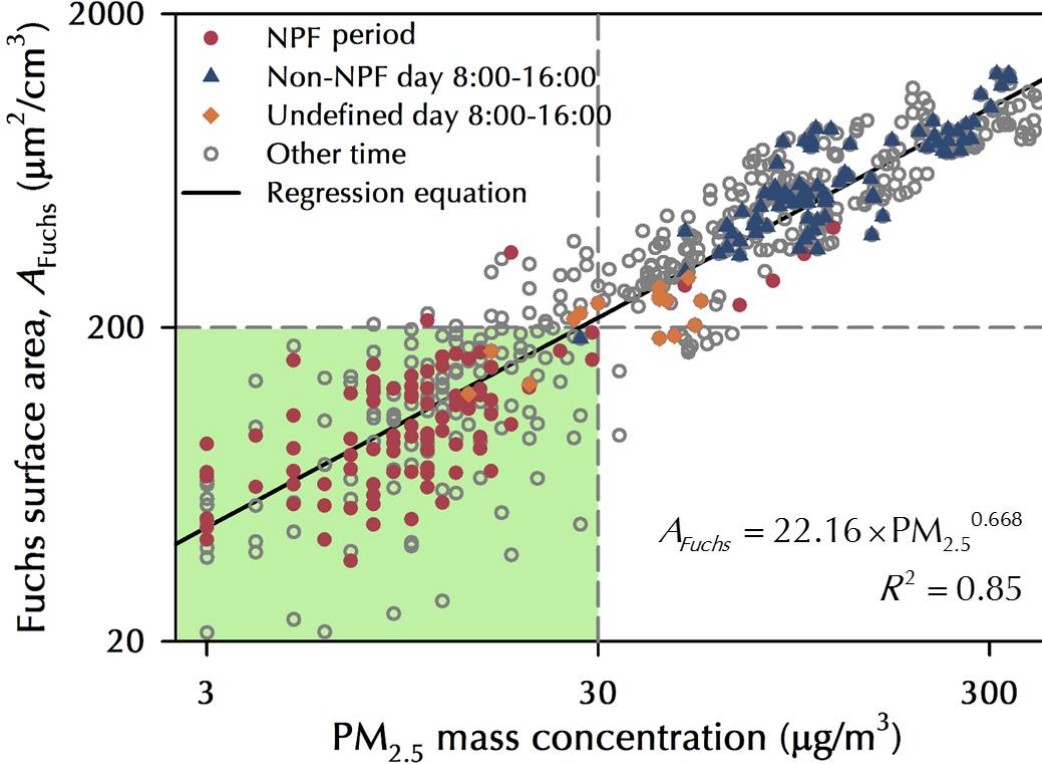


**Figure 11: Relationship between hourly averaged $A_{Fuchs}$ and the PM$_{2.5}$ mass concentration in Beijing. Data when $A_{Fuchs}$ changed rapidly was not included to avoid potential influence caused by the distance between Wanliu station and our campaign site. NPF period, daytime (8:00-16:00) on non-event days and undefined days, and other time are shown as different symbols. The regression of $A_{Fuchs}$ versus the PM$_{2.5}$ mass concentration was based on all the data. The proposed criterion for the occurrence**

**of NPF events, i.e., $A_{Fuchs}$ is lower than 200 μm$^2$/cm$^3$ (the PM$_{2.5}$ mass concentration is lower than 30 μg/cm$^3$), is shadowed by light green background.**