# Peer review of "Aerosol Surface Area Concentration: a Governing Factor for New Particle Formation in Beijing"

_Atmospheric Chemistry and Physics, 2017_

## Referee Comment (RC1) · Anonymous Referee #1 · 3 Jul 2017

This is a solid scientific analysis on factors influencing atmospheric NPF in Beijing. It is a pity that the analyzed data set is rather short, since the obtained results would probably be much more robust for a larger set of observed cases. On the other hand, it is understandable that comprehensive atmospheric observations do not usually cover long time periods. I have a number of minor comments to consider before recommending acceptance of this paper for publication in ACP.

Lines 50-55. The discussion here is a bit confusing. I suppose that it is meant to say that the observed values of AFuchs and Gamma lie is certain ranges.

Line 129: The parameter discussed here is usually called a sticking probability, not a coagulation efficiency.

Line 158.: I think "participate in" is a wrong wording here. Please modify.

Lines 216-217: The authors should explicitly define what is divided by what here.

Line 219: the project name should be EUCAARI

Lines 243-247: This part of the text is unclear and requires modification. First, it should be Figure 7, not 10. Second, the figure does not reveal governing factor determining the occurrence of NPF, but rather the area defined by two parameters affecting the NPF frequency in different locations. Third, by looking at figure I cannot agree that Afuchs lies in a narrow range in Hyytiala (not much narrower than in Beijing).

Lines 265-266: I thing this holds most of the atmospheric environments, being not specific to Beijing. I am not sure this statement is worth keeping here.

Line 270: The authors should be more specific in what they mean by the failure of Lgamma. I guess they mean that based on the values of Lgamma, no NPF event would have been expected to occur.

Line 311: This should be Figure 11, not 10.

The paper requires grammatical corrections. I recommend the authors to carefully check out the language with a native English speaker before submitting the revised version. Below is a list of some of the grammatical issues I notice when reading the paper:

L12: The analysis

L 20: A positive. . .

L 22: concentrations on NPF days were not . . .than those. . . varied

L 24: A good correlation

L 25: . . . to initial nucleation

L 32-34: concentrations. . .are formed. . .NPF events. . . .field observations. . .nuclei

sizes

L 41: of a NPF event

L72: . . .distributions have not. . .China, except for..

L79: The data analysis

L 80: The correlation

L 107: . . .observation compared with the previous

L 111: The PM2.5

L 123: A possibility

L 125: when an aerosol

L 133: The condensation sink

L 135: Since the condensation

L 143 and 145: change ; into ,

L 147: where GR is . . .

L 171: A total of 26 . . .day was

L 173: A day was

L 174 days were

L 181: predicted the occurrence of NPF. . .value of this parameter.

L 196: A positive. . .between the estimated

L 206: the correlation

L 207-216: several articles are missing from the text on these lines

L 227: is reasonable

L 241: environments

L 282: remained at a relatively

L 288: the sulfuric acid

L 295: on the particle size. . .while the particle mass

L 297: particle ranging from. . .were the major

L 301: the particle. . .the number

L 302: was a good

L 303: level was. . .of the influence

L 319: The new particle. . .

---

## Referee Comment (RC2) · Anonymous Referee #2 · 26 Jul 2017

**Summary:**

This work elucidated the connections between aerosol Fuchs surface area and NPF events in polluted megacity. The simultaneous measurements of particle size distribution down to ~1 nm and gaseous sulfuric acid concentration were firstly conducted in Beijing. The manuscript fits well to the scope of ACP and presents valuable results. Thus I recommend it to be published after the minor comments.

**Comments:**

1. In the abstract (page 1, lines 15-16): It seems that this sentence is not complete. Please revise.

2. Line 25: "It appears that the abundance of gaseous precursors such as sulfuric acid in Beijing is high enough to have nucleation", where does this conclusion come from? Do you mean that the measured sulfuric acid concentration is comparable with that in other places where NPF events were frequently observed? Please clarify. Have you considered other precursors?

3. Line 50-51: Please give a specific value/range.

4. Line 61: What do you mean transport? Do you indicate air mass origin, such as the air masses from the south direction are always associated with polluted situation in Beijing (Wehner et al., 2008; Wang et al., 2013a)?

5. Line 66-77: Gaseous sulfuric acid concentration was also measured in PRD, China (Wang et al., 2013b). Please add this reference.

6. Lines 179-180: See comments 4.

7. Lines 188-189: Please provide standard deviation.

8. Line 199: what do you mean NPF period, from 8:00-16:00?

9. Section 4.1: I would suggest to calculate the nucleation coefficients for the activation ($J_{1.5}=A$ {$H_2SO_4$}) and kinetic ($J_{1.5}=K$ {$H_2SO_4$}$^2$) nucleation mechanisms. This is very useful for the modeling study.

10. Line 233: For $A_{\text{Fuchs}} = 200$ $\mu m^2/cm^3$, could you also calculate the corresponding CS value?

11. Lines 301-302: I do not understand why the correlation is influenced by the nucleated particles. Do you mean the different particle number size distributions between nucleation and non-event days?

12. Section 4.2: It seems to me that you did not mention Figure 7 in the text. Please add one paragraph to explain it.

13. Line 316: 5 Conclusions

14. In Fig.1: Please use different colors to indicate non-event and undefined days.

15. In Figures 2 and 9: The unit of $A_{Fuchs}$ should be $\mu m^2/cm^3$.

16. Please check the language and the plots, especially for the mistakes in the writing and symbols.

**References**

Wang, Z. B., Hu, M., Wu, Z. J., Yue, D. L., He, L. Y., Huang, X. F., Liu, X. G., and Wiedensohler, A.: Long-term measurements of particle number size distributions and the relationships with air mass history and source apportionment in the summer of Beijing, Atmos Chem Phys, 13, 10159-10170, 10.5194/acp-13-10159-2013, 2013a.

Wang, Z. B., Hu, M., Yue, D. L., He, L. Y., Huang, X. F., Yang, Q., Zheng, J., Zhang, R. Y., and Zhang, Y. H.: New particle formation in the presence of a strong biomass burning episode at a downwind rural site in PRD, China, Tellus B, 65, 19965, http://dx.doi.org/10.3402/tellusb.v65i0.19965, 2013b.

Wehner, B., Birmili, W., Ditas, F., Wu, Z., Hu, M., Liu, X., Mao, J., Sugimoto, N., and Wiedensohler, A.: Relationships between submicrometer particulate air pollution and air mass history in Beijing, China, 2004-2006, Atmos Chem Phys, 8, 6155-6168, 2008.

---

## Author Comment (AC1) · 21 Aug 2017

We thank the reviewers for their helpful comments that have improved this manuscript. We have addressed the comments in the following paragraphs and have revised the revised manuscript correspondingly. Comments are shown as blue italic text followed by our responses. Changes are highlighted in the revised manuscript and shown as bold text in the responses. Line numbers quoted in the following responses correspond to the tracked change version of manuscript.

**Reviewer #1:**

*Comment*:

*This is a solid scientific analysis on factors influencing atmospheric NPF in Beijing. It is a pity that the analyzed data set is rather short, since the obtained results would probably be much more robust for a larger set of observed cases. On the other hand, it is understandable that comprehensive atmospheric observations do not usually cover long time periods. I have a number of minor comments to consider before recommending acceptance of this paper for publication in ACP.*

*1) Lines 50-55. The discussion here is a bit confusing. I suppose that it is meant to say that the observed values of AFuchs and Gamma lie is certain ranges.*

Response: To clarify the discussion, this paragraph has been revised as (lines 52-57) "**The values of $A_{Fuchs}$, however, were usually reported within a narrow range at locations such as Tecamac, Atlanta and Boulder (Kuang et al., 2010). Sulfuric acid concentration in Atlanta and Hyytiälä can differ significantly among days (Eisele et al., 2006; Petäjä et al., 2009). Therefore, sulfuric acid often governs nucleation and subsequent growth in the sulfur-rich atmosphere such as in Atlanta (McMurry et al., 2005). The growth enhancement factor, Γ, at Hyytiälä varied in a wide range while those at Tecamac and Boulder were found in a relatively narrow range.**"

*2) Line 129: The parameter discussed here is usually called a sticking probability, not a coagulation efficiency.*

Response: This parameter has been revised as "**mass accommodation coefficient (sticking efficiency)**" in line 134.

*3) Line 158.: I think "participate in" is a wrong wording here. Please modify.*

Response: The sentence was revised as "**Note that in Eq.(2) the absolute sulfuric acid concentrations were effectively normalized by the corresponding daily sulfuric acid maximum concentrations and thus has no influence on $L_\Gamma$ values and conclusions based on $L_\Gamma$ reported in this study.**" in lines 159-161.

*4) Lines 216-217: The authors should explicitly define what is divided by what here.*

Response: The sentence was revised as "**Estimated $\Gamma$ value for each event was normalized by the geometric mean $\Gamma$ value for the whole campaign to make it comparable with those obtained from previous studies**" in lines 217-219.

*5) Line 219: the project name should be EUCAARI*

Response: It has been corrected in lines 221 and 227.

*6) Line 243-247: This part of the text is unclear and requires modification. First, it should be Figure 7, not 10. Second, the figure does not reveal governing factor determining the occurrence of NPF, but rather the area defined by two parameters affecting the NPF frequency in different locations. Third, by looking at figure I cannot agree that $A_{Fuchs}$ lies in a narrow range in Hyytiala (not much narrower than in Beijing).*

Response: The figure label has been corrected accordingly. "Hyytiälä" was a typo and it was meant to be "Boulder". It was revised as "**The variations of these parameters at various locations are illustrated in Fig. 7. In Atlanta and Boulder, $A_{Fuchs}$ values fluctuated within relatively narrow ranges while the concentrations of gaseous precursors participating in nucleation differed significantly. The variations of $L_\Gamma$ at these locations were mainly caused by the relatively large variations in the concentrations of gaseous precursors. However, the contribution of gaseous precursors to $L_\Gamma$ in Beijing was relatively stable, and the variations of $L_\Gamma$ were mainly caused by the variations in $A_{Fuchs}$ values.**" (lines 248-252)

*7) Lines 265-266: I thing this holds most of the atmospheric environments, being not specific to Beijing. I am not sure this statement is worth keeping here.*

Response: This statement has been removed from the manuscript.

*8) Line 270: The authors should be more specific in what they mean by the failure of $L_\Gamma$. I guess they mean that based on the values of $L_\Gamma$, no NPF event would have been expected to occur.*

Response: It was revised as "**As indicated by Table 1, this exception was caused by the failure of $L_\Gamma$ rather than $A_{Fuchs}$ alone, i.e., NPF events occurred when estimated $L_\Gamma$ was greater than unity (the empirical threshold value).**" (Lines 270-272)

Response: They have been corrected accordingly.

*10) The paper requires grammatical corrections. I recommend the authors to carefully check out the language with a native English speaker before submitting the revised version. Below is a list of some of the grammatical issues I notice when reading the paper: L 12……L 227.*

Response: As suggested by the reviewer, we have revised the manuscript again to correct grammar errors.

**Reviewer #2:**

*Comment*:

*This work elucidated the connections between aerosol Fuchs surface area and NPF events in polluted megacity. The simultaneous measurements of particle size distribution down to ~1 nm and gaseous sulfuric acid concentration were firstly conducted in Beijing. The manuscript fits well to the scope of ACP and presents valuable results. Thus I recommend it to be published after the minor comments.*

*1) In the abstract (page 1, lines 15-16): It seems that this sentence is not complete. Please revise.*

Response: It has been revised as "**A dimensionless factor, $L_\Gamma$, characterized by the relative ratio of the coagulation scavenging rate over the condensational growth rate (Kuang et al., 2010), was applied in this work to reveal the governing factors for NPF events in Beijing.**"

*2) Line 25: "It appears that the abundance of gaseous precursors such as sulfuric acid in Beijing is high enough to have nucleation", where does this conclusion come from? Do you mean that the measured sulfuric acid concentration is comparable with that in other places where NPF events were frequently observed? Please clarify. Have you considered other precursors?*

Response: This statement was based on the observations that the maximum sulfuric acid concentrations on NPF days were not significantly higher (even lower, sometime) than those on non-event days. The reason that no NPF events were observed was because newly formed particles (and clusters) were quickly scavenged due to coagulation before they could continue to grow larger. To clarify this statement, we have revised the statement in Lines 22-27 as: **"However, the maximum sulfuric acid concentrations on NPF days were not significantly higher (even lower, sometime) than those on non-event days, indicating that the abundance of sulfuric acid in Beijing was high enough to initiate nucleation, but may not necessarily lead into NPF events. Instead, $A_{Fuchs}$ in Beijing varied greatly among days with a geometric standard deviation of 2.56, while that in Tecamac, Atlanta, and Boulder were reported to be much less variable. In addition, there was a good correlation between $A_{Fuchs}$ and $L_\Gamma$ in Beijing ($R^2 = 0.88$). Therefore, it was $A_{Fuchs}$ that fundamentally determined the occurrence of NPF events."**

*3) Line 50-51: Please give a specific value/range.*

Response: We have inserted the following statement "(**e.g., from several thousand into ~$1.5 \times 10^6$ #/cm$^3$**

**in this campaign)**" in lines 51-52.

*4) Line 61: What do you mean transport? Do you indicate air mass origin, such as the air masses from the south direction are always associated with polluted situation in Beijing (Wehner et al., 2008; Wang et al., 2013a)?*

*& 6) Lines 179-180: See comments 4.*

Response: It referred to the air mass origin. We revised it as "changes in air mass origins" and these references have been added in line 63 and line 181, respectively.

*5) Line 66-77: Gaseous sulfuric acid concentration was also measured in PRD, China (Wang et al., 2013b). Please add this reference.*

Response: The following sentence has been added: "**The same instrument used in the Beijing campaign was also deployed in Kaiping to measure sulfuric acid concentration during a one-month campaign in 2008 (Wang et al., 2013a).**"

*7) Lines 188-189: Please provide standard deviation.*

Response: We have revised the sentence as "In this campaign **(see Table 1)**, the median and mean values of $L_\Gamma$ on NPF days were 0.55 and 0.71 **(with a standard deviation of 0.40)**, respectively, comparing to 3.05 and 3.45 on non-event days **(with a standard deviation of 1.79)**, respectively."

*8) Line 199: what do you mean NPF period, from 8:00-16:00?*

Response: The starting time and ending time of NPF events varied among all event days. In this study, the NPF period was determined by the criterion that the estimated $J_{1.5}$ was greater than zero. We have revised lines 197-198 as:

"**There was a positive correlation between the estimated new particle formation rate, $J_{1.5}$, and sulfuric acid concentration during most NPF periods (typically 8:00-16:00 when the estimated $J_{1.5}$ was greater than zero).**"

*9) Section 4.1: I would suggest to calculate the nucleation coefficients for the activation ($J_{1.5}=A\cdot[H_2SO_4]$) and kinetic ($J_{1.5}=K\cdot[H_2SO_4]^2$) nucleation mechanisms. This is very useful for the modeling study.*

Response: The mean nucleation coefficients for the activation mechanism ($A$) and the kinetic mechanism ($K$) have been evaluated as $39.8 \times 10^{-6}$ s$^{-1}$ and $54.4 \times 10^{-12}$ cm$^3$ s$^{-1}$,respectively. These values of fitted nucleation coefficients are greater than those in a previous study (Wang et al., 2011). The HR-TOF-CIMS was calibrated everyday and background checks were performed each hour during daytime. The observed comparatively low sulfuric acid concentration might be due to relatively weak solar radiation intensity encountered in this springtime observation or other potential losses (lines 109-113). Future studies may help to resolve this.

*10. Line 233: For $A_{\text{Fuchs}}$=200 μm$^2$/cm$^3$, could you also calculate the corresponding CS value?*

Response: The relationship between CS and $A_{\text{Fuchs}}$ is shown as,

$$A_{\text{Fuchs}} = \frac{4\lambda}{3D} \cdot \text{CS}$$

where $\lambda$ is the mean free path, and $D$ is the diffusion coefficient of sulfuric acid (or hydrated sulfuric acid). Since different diffusion coefficient values are used in various studies, we prefer to reporting $A_{\text{Fuchs}}$ to prevent potential confusions. For $A_{\text{Fuchs}} = 200$ μm$^2$/cm$^3$, the corresponding CS value is ~0.27 s$^{-1}$ if assuming that sulfuric acid diffusion coefficient is 0.117 cm$^{-2}$s$^{-1}$.

*11. Lines 301-302: I do not understand why the correlation is influenced by the nucleated particles. Do you mean the different particle number size distributions between nucleation and non-event days?*

Response: Yes, the correlation between $A_{\text{Fuchs}}$ and the PM$_{2.5}$ mass concentration is determined by ambient particle size distribution. On NPF days, a high concentration of newly formed particles can lead to a rapid increase in $A_{\text{Fuchs}}$, while pre-existing aerosols are usually less abundant compared to those on non-event days. New particle formation changes the shape of ambient particle size distribution, or more intuitively, changes the shape of d$\overline{A_{\text{Fuchs}}}$/dlog$d_p$ (as shown in Fig. 10). We have revised lines 300-304 as "Figure 11 illustrates a good correlation between $A_{\text{Fuchs}}$ and the PM$_{2.5}$ mass concentration in Beijing with a $R^2$ of 0.85, although the correlation at low $A_{\text{Fuchs}}$ level was not as good as that at high $A_{\text{Fuchs}}$ level because **particles formed by nucleation significantly changed the shape of particle size distribution functions on NPF days.**"

*12. Section 4.2: It seems to me that you did not mention Figure 7 in the text. Please add one paragraph*

*to explain it.*

Response: Fig. 7 has been mislabeled as "Fig. 10" in the original manuscript. We are sorry for the confusion. The portion of the manuscript to describe the Fig. 7 lies in lines 248-256 as:

"The variations of $L_\Gamma$ at these locations were mainly caused by the relatively large variations in the concentrations of gaseous precursors. However, the contribution of gaseous precursors to $L_\Gamma$ in Beijing was relatively stable, and the variations of $L_\Gamma$ were mainly caused by the variations in $A_{\text{Fuchs}}$ values.

The predominant role of $A_{\text{Fuchs}}$ in Beijing can also be explained by using the balance formula shown as Eq. (4). It is dN/dt rather than the formation rate, $J$, that directly reflects whether a NPF event has occurred or not. dN/dt is the balanced result of the formation rate and the net *CoagSnk*. Different from $L_\Gamma$ that is the ratio of the particle loss rate over the growth rate, the ratio of the net *CoagSnk* over $J$ represents how many nucleated particles are lost due to the coagulation scavenging."

*13. Line 316: 5 Conclusions*

Response: It has been corrected accordingly.

*14. In Fig.1: Please use different colors to indicate non-event and undefined days.*

Response: The colors have been changed, and the corresponding descriptions have been revised as "**Identified thirteen non-event days and two undefined days are shadowed by grey and yellow backgrounds, respectively.**"

*15. In Figures 2 and 9: The unit of A Fuchs should be* $\mu m^2/cm^3$.

Response: It has been corrected accordingly.

*16. Please check the language and the plots, especially for the mistakes in the writing and symbols.*

Response: As suggested by the reviewer, we have revised the manuscript thoroughly.

---

## Author Response (AR2)

**Responses to Editor's Comments on Manuscript acp-2017-467**

**(Aerosol Surface Area Concentration: a Governing Factor for New Particle Formation in Beijing)**

We appreciate the comments and suggestions from the editor.   They were addressed here and revisions of the manuscript were made accordingly.

*Comment:*

*Concerning the comment 19 by ref 2, I think it would be worth adding the value of CS corresponding to the given $A_{Fuchs}$ threshold value of 200 $um^2/cm^3$ on line 235 (for example, adding "CS= NN $s^{-1}$" into parenthesis there; note the preferred unit of $s^{-1}$, not $cm^{-2}s^{-1}$ in current literature). This is because most other papers report values of CS, not those of $A_{Fuchs}$, these days. When discussing further the $A_{Fuchs}$ (or CS) threshold observed here, it might be beneficial to add couple of sentences on whether such threshold has been observed anywhere else (e.g. Salma et al 2016, ACP page 8715-28 found such threshold in Budabest, Hungary, but at a particle loading of CS = 0.02 $s^{-1}$).*

Response: We added "**(the corresponding condensation sink is 0.027 $s^{-1}$)**" in line 235 and "**A similar threshold (the condensation sink of 0.02 $s^{-1}$) was found in Budapest, Hungary (Salma et al., 2017)**" in lines 239-240.

*Comments:*
*diurnal sulfuric acid -- > the diurnal sulfuric acid concentration, sulfuric acid concentration -- > the sulfuric acid concentration,*
*sulfuric acid -- > sulfuric acid concentration (lines 50, 54, 64, 70, 84, 154, 287, 321)*
*Abstract, sections 2, 4 and 5: Please check out the format of presenting dates in ACP.*
*line 19: environments, such as in Boulder and Hyytiälä, the dayly-maximum…*
*line 25: 2.56, whereas the variabilities of AFuchs in…*
*line 77: a good correlation*
*line 97: please specify the size. ... 10 um in particle aerodynamic diameter. ???*
*line 100: The particle density…*
*lines 120-130: I think this paragraphs would benefit from 1-3 references.*
*line 157: are -- > represent*
*line 178: the sub-3 nm…*
*line 183: This sentence is unclear, please modify.*
*line 185: ...were assumed to be…*
*line 194, The following…*
*lines 205-206: However, this exponent is ... from the exponents no greater…., indicating that activation or…*
*Line 215: what is the meaning of "p=1" here?*
*line 223: in measured sulfuric acid concentrations…*
*line 233: NPF days, whereas…*
*line 243: delete "the" before R2*
*line 265: Fig. 8), indicating high coagulation losses…*

*line 267: rather than were grown into larger sizes, such that…*

*line 287: led to…particles were*

Response: We appreciated detailed editing from the editor that helps to improve this manuscript. We have made the suggested changes. In addition, we read the manuscript again to correct grammar errors.